# Progression from external pilot to definitive randomised controlled trial: a methodological review of progression criteria reporting

Katie Mellor [iD] ,[1] Saskia Eddy,[2] Nicholas Peckham,[1] Christine M Bond,[3] Michael J Campbell [iD] ,[4] Gillian A Lancaster,[5] Lehana Thabane,[6] Sandra M Eldridge,[2] Susan J Dutton,[1] Sally Hopewell[1]

**Correspondence to**
Katie Mellor;
katie.mellor@ndorms.ox.ac.uk

## ABSTRACT

**Objectives** Prespecified progression criteria can inform the decision to progress from an external randomised pilot trial to a definitive randomised controlled trial. We assessed the characteristics of progression criteria reported in external randomised pilot trial protocols and results publications, including whether progression criteria were specified a priori and mentioned in prepublication peer reviewer reports.

**Study design** Methodological review.

**Methods** We searched four journals through PubMed: *British Medical Journal Open*, *Pilot and Feasibility Studies*, *Trials* and *Public Library of Science One*. Eligible publications reported external randomised pilot trial protocols or results, were published between January 2018 and December 2019 and reported progression criteria. We double data extracted 25% of the included publications. Here we report the progression criteria characteristics.

**Results** We included 160 publications (123 protocols and 37 completed trials). Recruitment and retention were the most frequent indicators contributing to progression criteria. Progression criteria were mostly reported as distinct thresholds (eg, achieving a specific target; 133/160, 83%). Less than a third of the planned and completed pilot trials that included qualitative research reported how these findings would contribute towards progression criteria (34/108, 31%). The publications seldom stated who established the progression criteria (12/160, 7.5%) or provided rationale or justification for progression criteria (44/160, 28%). Most completed pilot trials reported the intention to proceed to a definitive trial (30/37, 81%), but less than half strictly met all of their progression criteria (17/37, 46%). Prepublication peer reviewer reports were available for 153/160 publications (96%). Peer reviewer reports for 86/153 (56%) publications mentioned progression criteria, with peer reviewers of 35 publications commenting that progression criteria appeared not to be specified.

**Conclusions** Many external randomised pilot trial publications did not adequately report or propose prespecified progression criteria to inform whether to proceed to a future definitive randomised controlled trial.

## Strengths and limitations of this study

► We conducted a large recent assessment of the use and reporting of progression criteria in publications reporting external randomised pilot trial protocols and results.

► As this study only investigated external randomised pilot trial publications, it is unclear whether the findings can be generalised to other external feasibility study designs such as non-randomised pilot trials and non-pilot feasibility studies.

► One researcher independently screened all publications, assessed eligibility and extracted data from all included publications, while other members of the research team provided a second data extraction for 25% of the included publications.

## INTRODUCTION

Pilot trials aim to determine whether a future definitive randomised controlled trial (RCT) is feasible.[1] Internal pilot trials are embedded in the RCT design forming its first phase.[2] In contrast, external pilot trials are small stand-alone studies conducted before a definitive RCT. Prespecified progression criteria can help researchers interpret the findings of an external randomised pilot to decide whether the future definitive RCT is or is not feasible, and whether changes should be made to the trial design. Progression criteria should be specified before the pilot trial begins (a priori) to avoid introducing bias associated with establishing progression criteria once external pilot trial findings are known.

A 2019 review found that less than 20% of randomised pilot trial protocols published between 2013 and 2017 reported clear progression criteria. Trial features, such as a more recent publication year and certain countries of origin, were associated with reporting progression criteria.[3] The 2016

Consolidated Standards of Reporting Trials (CONSORT) extension for reporting randomised pilot and feasibility trials advises that 'at a minimum there should be something reported to suggest how the decision to progress to the definitive study will be made'.[4] The extent to which this guidance has improved progression criteria reporting in more recently published pilot trials is unclear. Although previous research has investigated whether progression criteria are reported, the quality of progression criteria reporting—including how the criteria are established during pilot trial design and assessed on pilot trial completion—has not yet been investigated.

We conducted a methodological review to investigate the application and reporting of progression criteria in a recent sample of external randomised pilot trial protocol and results publications. The primary objective was to describe the reporting of progression criteria, including the areas of feasibility that progression criteria were based on as described in a published framework of reasons for conducting pilot trials,[5] their rationale or justification and who established and assessed the progression criteria. One set of secondary objectives were to assess whether the progression criteria reported in pilot trial results publications were specified a priori in a published protocol or trial registration and whether the results publication reported the intention to progress to a definitive RCT. We also assessed the extent and context in which progression criteria were discussed in prepublication peer reviewer reports where available for the protocol and results publications.

## METHODS

### Protocol and registration

A protocol for this research is registered on the Open Science Framework: osf.io/bn35k.[6] A summary of the methods used is detailed below. This review is reported following the Preferred Reporting Items for Systematic Reviews and Meta-Analyses (PRISMA) statement.[7]

### Eligibility criteria

We included all protocol and results publications for external randomised pilot trials that reported progression criteria and were published between January 2018 and December 2019 inclusive. Progression criteria were defined as criteria to inform the decision to progress to a definitive RCT. Included publications were published in the English language and were not restricted by intervention, health-related context or setting.

### Information sources

Four journals were searched through PubMed: *British Medical Journal (BMJ) Open*, *Pilot and Feasibility Studies (PAFS)*, *Trials* and *Public Library of Science (PLoS) One*. These journals were chosen because they are known to publish pilot trial protocol and results publications and had published the most PubMed indexed publications that included the terms 'pilot' or 'feasibility and 'trial' or

'protocol' in their title within 2018 and 2019. All included journals direct authors to the CONSORT statement[8] reporting guideline: *BMJ Open* and *Trials* advise authors to use the most appropriate statement extension, and the *PAFS* journal directs authors to the CONSORT Extension to Pilot and Feasibility Trials.[4]

Search terms included 'pilot' or 'feasibility' in the title, and 'trial', 'study' or 'protocol' in the title or abstract. See online supplemental file 1 for the full search strategy which was last used on 6 January 2020.

### Study selection

Titles and abstracts of identified publications were screened against inclusion criteria. Full texts were retrieved for those that appeared relevant and screened against a predefined eligibility checklist by KM. All included publications were saved in EndNote V.X9 for Windows. Where both the protocol and corresponding pilot trial result publication were identified, both were included.

### Data collection

Data extraction forms produced in Microsoft Excel (Office 16) were prepiloted on the first 10 trials ordered alphabetically to ensure usability and completeness (the data extraction form used can be obtained from osf.io/fxv4n). One researcher (KM) extracted the data for all included publications. Other team members (SEd and NP) conducted a second data extraction for a randomly selected 25% sample. As we found minimal differences between the two data extractions, we decided not to conduct double data extraction for all of the included publications.

From trial protocol and results publications, we extracted: trial characteristics (including author, year, journal, country, randomisation design, therapeutic area, intervention type, sample size target, number of arms and single or multicentre); feasibility objectives, outcomes and instances of hypothesis testing; progression criteria details (wording, rationale or justification, format, process for establishing and process for assessing); and references to progression criteria in prepublication peer reviewer reports, where these were published online and linked to the publication.

For completed pilot trial results publications, we also extracted: whether progression criteria were met; any reported intention to progress to a definitive RCT; any proposed changes to the definitive RCT design; any refinement of hypotheses; any comment on data quality; and whether progression criteria had changed from the corresponding protocol or trial registration publication, if a published protocol was not available.

### Synthesis of results

Descriptive statistics (frequencies and the mean, median and IQR for trial sample sizes) were produced to describe trial characteristics and address our primary and secondary objectives. Data were analysed using Stata

V.15.0 (StataCorp).[9] We report the frequency with which different feasibility uncertainties contributed to progression criteria using prespecified domains of reasons for conducting pilot trials: process, resource, management and scientific.[5] The mean number of progression criteria specified per trial was also calculated.

We used narrative synthesis to describe the context in which progression criteria were mentioned in publicly available prepublication peer reviewer reports (synthesised by KM). We did not use a predefined checklist to formally assess peer reviewer reports and we do not comment on the quality of peer review. Instead, we simply looked for any mention that progression criteria were not present in the prepublication manuscript, and any queries about rationale for progression criteria used.

We did not aim to comment on the quality of the evidence from the studied randomised pilot trials. We aimed to comment only on the quality of reporting of progression criteria in this sample of external randomised pilot trial protocol and results publications.

## Patient and public involvement

Patients or the public were not involved in the design, conduct, reporting or dissemination plans of this research.

## RESULTS
### Study selection

Our search strategy identified 1030 publications. We screened their titles and abstracts, then assessed the full texts of 679 publications for eligibility. One hundred and sixty publications were eligible for our study. Figure 1 shows the full PRISMA flow chart of publications included

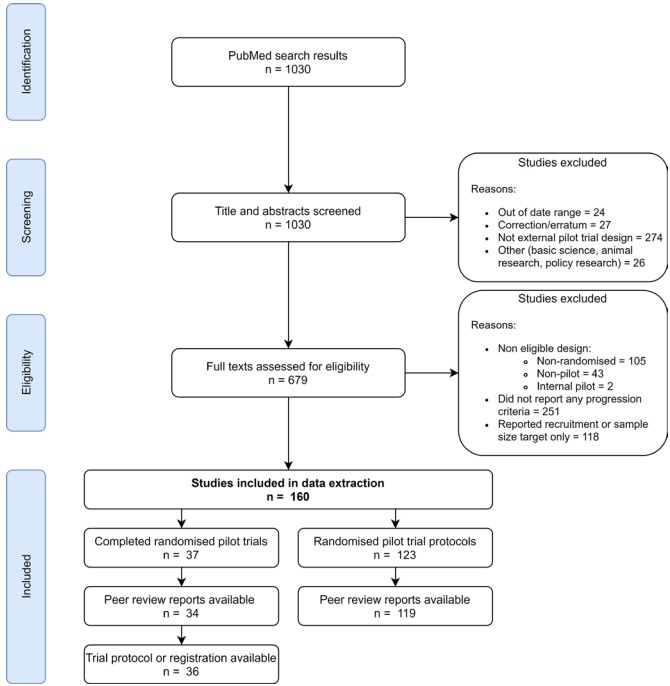

**Figure 1** Preferred Reporting Items for Systematic Reviews and Meta-Analyses (PRISMA) flow chart.

and excluded at each stage. We excluded many publications during full text screening as we were unable to identify explicit progression criteria (n=251), or where publications only reported a recruitment or sample size target (n=118). Online supplemental file 2 lists the included publications describing external pilot trial protocols and results. We found two instances where both the completed trial publication and protocol were identified. In these instances, both were included.

### Study characteristics

Table 1 summarises the characteristics of the included publications. Most of the publications were pilot trial protocols (123/160, 77%) rather than completed pilot trial results (37/160, 23%). The journal with the most eligible publications was *PAFS* (77/160, 48%). Most publications described external pilot trials that were two-arm (143/160, 89%), multicentre (102/160, 64%), non-industry-funded (147/160, 92%) trials of counselling, lifestyle or physiotherapy interventions (94/160, 59%). The reported trials covered 27 therapeutic areas and trials were from 18 countries, mostly from the UK (87/160, 54%).

Primary feasibility objectives were explicitly stated in 71/160 (44%) publications, and most publications reported feasibility outcomes in the methods that addressed all of the stated feasibility objectives (109/160, 68%). In 50/160 (31%) of the publications, the stated feasibility outcomes only somewhat addressed trial objectives, often because the objective stated was broad (eg, 'to determine whether a future trial is feasible') and did not define specific aspects of feasibility being assessed. With respect to data collection and assessment of feasibility outcomes, completely defined prespecified assessments or measurements were often stated (140/160, 88%). Most of the pilot trial publications that reported the intention to conduct hypothesis testing stated that this was exploratory or advised caution in interpretation. All but one publication reported multiple feasibility outcomes. The place in the publication where the specific uncertainties related to trial feasibility were first reported varied, but most often this was within the pilot trial feasibility objectives (72/160, 45%), or within the data collection section describing the feasibility outcomes (26/160, 16%) or pilot trial assessments or measurements (23/160, 14%).

### Characteristics of progression criteria

Characteristics of progression criteria are presented in table 2. The reported progression criteria generally addressed some (99/160, 62%) or all (53/160, 33%) of the pilot trial's feasibility outcomes. The pilot trial publications reported a mean of 4 (mean 4.05) progression criteria targets per pilot trial. Recruitment (113/160, 71%) and retention (106/160, 66%) were the most commonly reported indicators of feasibility to inform progression. In total, we identified 58 distinct areas of trial feasibility that contributed to progression criteria, which we grouped into four domains: process, resource,

**Table 1** Characteristics of the studied external randomised pilot trial publications

| | Completed (n=37) n (%) | Protocol (n=123) n (%) | Total (n=160) n (%) |
|---|---|---|---|
| Journal | | | |
| *British Medical Journal (BMJ) Open* | 11 (30) | 34 (28) | 45 (28) |
| *Pilot and Feasibility Studies (PAFS)* | 21 (57) | 56 (46) | 77 (48) |
| *Trials* | 2 (5) | 33 (27) | 35 (22) |
| *Public Library of Science (PLoS) One* | 3 (8) | 0 (0) | 3 (2) |
| Country | | | |
| Australia | 6 (16) | 4 (3) | 10 (6) |
| Brazil | 0 (0) | 1 (1) | 1 (1) |
| Canada | 4 (11) | 15 (12) | 19 (12) |
| China | 0 (0) | 4 (3) | 4 (3) |
| Denmark | 0 (0) | 1 (1) | 1 (1) |
| Germany | 1 (3) | 1 (1) | 2 (1) |
| Korea | 0 (0) | 1 (1) | 1 (1) |
| Nepal | 1 (3) | 2 (2) | 3 (2) |
| New Zealand | 2 (5) | 1 (1) | 3 (2) |
| Norway | 1 (3) | 0 (0) | 1 (1) |
| Ireland | 0 (0) | 5 (4) | 5 (3) |
| Sweden | 1 (3) | 1 (1) | 2 (1) |
| Tanzania | 0 (0) | 1 (1) | 1 (1) |
| Thailand | 0 (0) | 1 (1) | 1 (1) |
| The Netherlands | 0 (0) | 2 (2) | 2 (1) |
| UK | 19 (51) | 68 (55) | 87 (54) |
| USA | 2 (5) | 14 (11) | 16 (10) |
| Zimbabwe | 0 (0) | 1 (1) | 1 (1) |
| Funder | | | |
| Industry | 2 (5) | 2 (2) | 4 (3) |
| Non-industry | 32 (86) | 115 (94) | 147 (92) |
| A combination | 1 (3) | 4 (3) | 5 (3) |
| Unknown | 2 (5) | 1 (1) | 3 (2) |
| Trial did not receive funding | 0 (0) | 1 (1) | 1 (1) |
| Therapeutic areas | | | |
| Complementary medicine | 0 (0) | 1 (1) | 1 (1) |
| Anaesthesia | 1 (3) | 1 (1) | 2 (1) |
| Cardiology | 0 (0) | 3 (2) | 3 (2) |
| Critical care | 1 (3) | 7 (6) | 8 (5) |
| Endocrinology | 0 (0) | 6 (5) | 6 (4) |
| Gastroenterology | 1 (3) | 3 (2) | 4 (3) |
| Geriatrics | 1 (3) | 4 (3) | 5 (3) |
| Hepatology | 0 (0) | 1 (1) | 1 (1) |
| Infectious diseases | 0 (0) | 3 (2) | 3 (2) |
| Musculoskeletal | 6 (16) | 4 (3) | 10 (6) |
| Nephrology | 0 (0) | 1 (1) | 1 (1) |
| Neurology | 3 (8) | 12 (10) | 15 (9) |
| Obstetrics/gynaecology | 2 (5) | 6 (5) | 8 (5) |

Continued

**Table 1** Continued

| | Completed (n=37) n (%) | Protocol (n=123) n (%) | Total (n=160) n (%) |
|---|---|---|---|
| Oncology | 4 (11) | 7 (6) | 11 (7) |
| Ophthalmology | 0 (0) | 1 (1) | 1 (1) |
| Orthopaedics | 2 (5) | 1 (1) | 3 (2) |
| Other | 2 (5) | 3 (2) | 5 (3) |
| Otolaryngology (ENT) | 2 (5) | 1 (1) | 3 (2) |
| Paediatrics | 2 (5) | 3 (2) | 5 (3) |
| Pain | 1 (3) | 0 (0) | 1 (1) |
| Palliative care | 0 (0) | 3 (2) | 3 (2) |
| Psychiatry/psychology | 2 (5) | 19 (15) | 21 (13) |
| Public health | 2 (5) | 15 (12) | 17 (11) |
| Respiratory | 0 (0) | 2 (2) | 2 (1) |
| Rheumatology | 0 (0) | 1 (1) | 1 (1) |
| Surgery | 3 (8) | 8 (7) | 11 (7) |
| Trauma | 2 (5) | 7 (6) | 9 (6) |
| Intervention type | | | |
| Drug | 4 (11) | 9 (7) | 13 (8) |
| Surgery/procedure | 6 (16) | 13 (11) | 19 (12) |
| Counselling/lifestyle/physiotherapy | 22 (59) | 72 (59) | 94 (59) |
| Equipment | 4 (11) | 5 (4) | 9 (6) |
| Other | 1 (3) | 24 (20) | 25 (16) |
| Sample size target* | | | |
| Mean (SD) | 72.8 (62.5) | 258.5 (1215.7) | 217.3 (1074.9) |
| Median (IQR) | 60 (32–90) | 60 (40–100) | 60 (40–100) |
| Min-Max | 6–300 | 20–12 000 | 6–12 000 |
| *Cluster randomised pilot trials* | *(n=3)* | *(n=18)* | *(n=21)* |
| Number of clusters | | | |
| Mean (SD) | 7.3 (2.3) | 9.7 (11.6) | 9.3 (10.7) |
| Median (IQR) | 6 (6–10) | 6 (3–10) | 6 (4–10) |
| Min-Max | 6–10 | 2–45 | 2–45 |
| Number of arms | | | |
| 2 | 32 (86) | 111 (90) | 143 (89) |
| >2 | 5 (14) | 12 (10) | 17 (11) |
| Number of centres | | | |
| Single centre | 19 (51) | 36 (29) | 55 (34) |
| Multicentre | 18 (49) | 84 (68) | 102 (64) |
| Unclear | 0 (0) | 3 (2) | 3 (2) |
| Feasibility objective/s explicitly described as primary | | | |
| Yes | 9 (24) | 62 (50) | 71 (44) |
| No | 28 (76) | 61 (50) | 89 (56) |
| Trial outcomes address trial objectives | | | |
| Yes | 18 (49) | 91 (74) | 109 (68) |
| No | 0 (0) | 1 (1) | 1 (1) |
| Somewhat† | 19 (51) | 31 (25) | 50 (31) |
| Completely defined prespecified assessments or measurements stated | | | |

**Table 1** Continued

|  | Completed (n=37) n (%) | Protocol (n=123) n (%) | Total (n=160) n (%) |
|---|---|---|---|
| Yes | 27 (73) | 113 (92) | 140 (88) |
| Not for every outcome | 10 (27) | 10 (8) | 20 (13) |
| Hypothesis testing |  |  |  |
| Yes | 2 (5) | 16 (13) | 18 (11) |
| Yes, exploratory/caution advised | 18 (49) | 43 (35) | 61 (38) |
| No | 17 (46) | 64 (52) | 81 (51) |
| Number of uncertainties reported |  |  |  |
| One | 0 (0) | 1 (1) | 1 (1) |
| Multiple | 37 (100) | 122 (99) | 159 (99) |
| Where uncertainties are first reported (excluding abstract) |  |  |  |
| Introduction | 0 (0) | 1 (1) | 1 (1) |
| Research question(s) | 2 (5) | 4 (3) | 6 (4) |
| Aim(s) | 5 (14) | 16 (13) | 21 (13) |
| Objective(s) | 10 (27) | 62 (50) | 72 (45) |
| Outcome(s) | 9 (24) | 17 (14) | 26 (16) |
| Outcome measure(s) | 7 (19) | 16 (13) | 23 (14) |
| Analysis | 0 (0) | 2 (2) | 2 (1) |
| Within the text under a feasibility/uncertainty heading | 2 (5) | 1 (1) | 3 (2) |
| Throughout the text, not in one specific area | 2 (5) | 4 (3) | 6 (4) |

Percentages may not sum up to 100 due to rounding.
*Where publications reported a sample size target range (eg, 12–16 participants), the lower bound of the target is included. A sample size target was not reported in two publications (both reporting completed pilot trials and including the actual number of recruited participants).
†Trial objective was vague (eg, to 'assess feasibility') and the specific areas of feasibility were not explicitly stated.
ENT, ear, nose and throat.

management and scientific. The domains and areas are listed in online supplemental file 3. Most of the areas were process uncertainties (34/58, 59%), which dealt with the feasibility of processes that are key to the success of the future definitive RCT.[5]

Four publications reported progression criteria that were based on detecting potential efficacy, including determining non-inferiority of the intervention compared with a comparator, determining intervention superiority at follow-up and finding a trend for difference between the intervention and comparator groups on clinical outcomes.

### Progression criteria and quantitative indicators of feasibility

All of the pilot trial protocol and result publications reported using quantitative indicators of trial feasibility (eg, rate of recruitment and amount of missing data) to inform at least one of the trial's progression criteria, with 78% (125/160) basing all progression criteria on quantitative indicators.

All but seven publications reported quantifiable numerical thresholds that were, or would be, used to assess the progression criteria. The seven remaining publications did not report specific quantifiable targets for progression

criteria, but did report how the decision to progress from pilot to definitive RCT would be made and the feasibility indicators that would be considered when making this decision.

The quantifiable numerical targets used were most often reported as a distinct threshold (eg, achieving a specified rate of recruitment, retention or data completion) (133/160, 83%). This was followed by a traffic light approach to reporting progression criteria (20/160, 13%) with thresholds correlating to different domains (eg, above a higher threshold (green) indicating the definitive trial is feasible/proceed, within a mid/acceptable threshold (amber) indicating that changes to definitive trial are required, and below a lower threshold (red) indicating that the definitive trial is not feasible/not proceed).

### Progression criteria and qualitative indicators of feasibility

Many publications reported planned or completed qualitative research as part of the randomised pilot trial (108/160, 68%). Although the findings from qualitative research conducted as part of a pilot trial are often reported in a separate publication, the intention to conduct qualitative research as part of a pilot trial should

**Table 2** Characteristics of progression criteria reported in external randomised pilot trial publications

| | Completed (n=37) n (%) | Protocol (n=123) n (%) | Total (n=160) n (%) |
|---|---|---|---|
| Feasibility outcomes informing progression criteria | | | |
| All | 14 (38) | 39 (32) | 53 (33) |
| Some | 22 (59) | 77 (63) | 99 (62) |
| None | 1 (3) | 2 (2) | 3 (2) |
| Unclear* | 0 (0) | 5 (4) | 5 (3) |
| **Reported process for establishing progression criteria** | | | |
| Who decided on progression criteria | | | |
| Reported | 4 (11) | 8 (6) | 12 (8) |
| Not reported | 33 (89) | 115 (94) | 148 (93) |
| Rationale for progression criteria | | | |
| Reported for all progression criteria | 8 (22) | 20 (16) | 28 (18) |
| Reported for some criteria only | 4 (11) | 12 (10) | 16 (10) |
| Not reported | 25 (68) | 91 (74) | 116 (73) |
| **Progression criteria format** | | | |
| Research method informing progression criteria | | | |
| Quantitative | 32 (86) | 93 (76) | 125 (78) |
| Quantitative and qualitative (mixed methods) | 5 (14) | 29 (24) | 34 (21) |
| Unclear | 0 (0) | 1 (1) | 1 (1) |
| Qualitative research contribution | | | |
| Informs progression criteria | 5 (14) | 29 (24) | 34 (21) |
| Does not inform progression criteria | 14 (38) | 60 (49) | 74 (46) |
| Qualitative research methodology not used | 18 (49) | 34 (28) | 52 (33) |
| Quantitative progression criteria target format | | | |
| Distinct threshold | 34 (92) | 99 (80) | 133 (83) |
| Traffic light system | 2 (5) | 18 (15) | 20 (13) |
| Other | 1 (3) | 6 (5) | 7 (4) |
| **Reported process for assessing progression criteria to inform the progression decision** | | | |
| Process for progression decision-making | | | |
| Reported | 16 (43) | 58 (47) | 74 (46) |
| Not reported | 21 (57) | 65 (53) | 86 (54) |
| Who is involved in assessing progression criteria | | | |
| Reported | 5 (14) | 30 (24) | 35 (22) |
| Not reported | 32 (86) | 93 (76) | 125 (78) |
| **Peer reviewer reports** | | | |
| Progression criteria mentioned in peer reviewer report | | | |
| Yes | 19 (51) | 67 (54) | 86 (54) |
| *Peer review comment theme* | | | |
| Progression criteria were not specified | 6 (32) | 29 (44) | 35 (41) |
| Unclear whether progression criteria were specified | 1 (5) | 4 (6) | 5 (6) |
| Progression criteria rationale or justification query | 5 (26) | 15 (22) | 20 (23) |
| Other | 7 (37) | 19 (29) | 26 (30) |
| No | 15 (41) | 52 (42) | 67 (42) |
| Peer reviewer report unavailable | 3 (8) | 4 (3) | 7 (4) |

Continued

| Table 2 Continued | | | |
|---|---|---|---|
| | Completed (n=37) n (%) | Protocol (n=123) n (%) | Total (n=160) n (%) |

Percentages may not sum up to 100 due to rounding.
*Feasibility uncertainties are not completely defined in the objectives and outcomes; key methodological uncertainties have been identified from those stipulated in the progression criteria.

be made explicit before the pilot trial commences and be included in the pilot trial protocol.[10] The intention to conduct qualitative research was reported in protocols (89/123, 72%) more often than the results of qualitative research were reported in pilot trial result publications (19/37, 51%). However, qualitative indicators of trial feasibility, such as participants or researchers' views of the acceptability of the trial or intervention collected in interviews, only informed progression criteria in 34 of the 108 (31%) publications that reported planned or completed qualitative research.

Two protocols reported multiple progression criteria for individual feasibility indicators, for example, reporting both a target for number of participants recruited, and another target for number of participants recruited in a given time frame. In one of these instances, the authors reported that all criteria would need to be met or met within reasonable limits (within the green or amber traffic light domain) to progress to a full trial without major study redesign. It was unclear in the other protocol whether meeting one criterion for each indicator of feasibility was sufficient justification for progression.

### Process for establishing progression criteria
Twelve pilot trial publications reported how the progression criteria had been established, with most involving a trial steering or oversight committee (10/12, 83%; five reported having patient or public representation), working with funders (3/12, 25%) and/or a trial management group (5/12, 42%; two reported having patient or public representation). Other examples included agreeing progression criteria with a data monitoring and ethics committee (1/12, 8%; reported having patient or public representation) or study physicians (1/12, 8%) or establishing progression criteria based on the author's clinical experience (1/12, 8%).

Forty-four of the 160 publications (28%) reported rationale or justification for all or some of the stated progression criteria. For 29 publications, the stated justification was previous related research, with 25 providing references to previous studies. Thirteen publications referenced sources of guidance and methodological research,[4 11–21] including three references to published guidance for internal pilot trials.[2] Four publications reported that contextual considerations had informed progression criteria (such as what would be an achievable recruitment rate, or intervention time frame in the definitive trial), and three reported that clinical considerations had informed criteria (including medical chart reviews, clinical advice and the nature of the population). Most of

the pilot trial publications (116/160, 73%) did not report any rationale or justification for choice of progression criteria.

### Process for assessing progression criteria
Nearly half of the publications (74/160, 46%) reported how progression criteria had or would inform the decision to progress to a future definitive RCT. This included whether changes to definitive RCT design would be considered if criteria were not strictly met (eg, were met within reasonable limits or within the aforementioned 'amber' traffic light range), or who was or would be involved in assessing progression criteria.

One publication reported a two-stage decision-making process with different criteria assessed at each stage. Stage 1 was to decide on the best intervention route, and stage 2 was to decide whether to take the optimal intervention route forward to a definitive RCT. Another publication described the intention to hold a consensus conference of key stakeholders (patients, surgeons, public representatives and researchers) to agree whether a definitive RCT was feasible. Four pilot trials referred to A Process for Decision-making after Pilot and Feasibility Trials framework[11] to facilitate progression decision-making.

Nearly a quarter of publications reported who would be involved in assessing progression criteria (35/160, 22%). An independent trial steering committee was most commonly involved (26/35, 74%). Other reported parties included the research team or trial management group (13/35, 37%), data monitoring committee (7/35, 20%), trial sponsor (2/35, 6%), funder (1/35, 3%), independent statistician (1/35, 3%) and other stakeholders, such as patients, clinicians and public representatives (3/35, 9%).

### Intentions of completed randomised pilot trial publications
Most completed pilot trials reported that a future RCT would be feasible or the intention to proceed (30/37, 81%), including the 17 completed pilot trials which met all of their progression criteria (table 3). Thirteen pilot trials met some of their progression criteria; of these, nine reported that a future RCT would be feasible, two reported that they would not proceed to a definitive RCT and two reported the intention to conduct further feasibility assessment. Four pilot trials did not meet their progression criteria, of which three reported that a future RCT would still be feasible with changes to study design. The extent to which progression criteria were met was unclear for three trials; of these, two reported the

**Table 3** Intentions reported in completed external randomised pilot trial results publications

| Data | Completed (n=37) n (%) |
|---|---|
| Progression criteria met | |
| All | 17 (46) |
| None | 4 (11) |
| Some | 13 (35) |
| Unclear | 3 (8) |
| Progression decision | |
| Proceed/future RCT is feasible | 30 (81) |
| *With intended design* | 0 (0) |
| *With amendments* | 28 (93) |
| *Not reported whether changes will be made to definitive RCT design* | 2 (7) |
| ***Funding intentions*** | |
| *Funding for definitive RCT identified* | 4 (13) |
| *Non-industry* | 3 (75) |
| *Unclear* | 1 (25) |
| *Expected funding for definitive RCT not reported* | 26 (87) |
| ***Timing intentions*** | |
| *Time frame of expected progression reported* | 1 (3) |
| *Time frame of expected progression not reported* | 29 (97) |
| Conduct further pilot/feasibility work | 4 (11) |
| Not proceed/future RCT is not feasible | 3 (8) |
| Justification reported for the progression decision reported | |
| Yes | 36 (97) |
| No | 1 (3) |
| Comment on data quality (eg, proportion of missing/incomplete data from questionnaires or results) | |
| Yes | 27 (73) |
| No | 10 (27) |
| Comment on refinement of hypotheses | |
| Yes | 1 (3) |
| No | 36 (97) |
| Published protocol available | |
| Yes | 16 (43) |
| No | 1 (3) |
| Alternative available (eg, trial registration or REC submission) | 20 (54) |
| Progression criteria in earlier trial record (protocol or registration) | |
| No change | 10 (28) |
| Yes change | 26 (72) |
| Reasons for change reported | 1 (4) |

Continued

**Table 3** Continued

| Data | Completed (n=37) n (%) |
|---|---|
| No reason for change reported | 3 (12) |
| Progression criteria were not reported in the earlier trial record | 22 (85) |

Percentages may not sum up to 100 due to rounding.
RCT, randomised controlled trial; REC, Research Ethics Committee.

intention to conduct further feasibility assessment, and one reported that a future RCT would be feasible.

All but two of the completed pilot trials that reported that a future RCT would be feasible planned to make changes to their definitive RCT design (28/30, 93%). Of these, four reported the implications of the pilot trial findings in a table format, alongside whether progression criteria had or had not been met. Proposed changes included altering eligibility criteria, recruitment strategies (eg, number of sites, recruitment materials, recruitment setting), randomisation design, blinding, outcome measures, follow-up schedules and duration, and seeking additional research team support (such as a dedicated trial manager, research coordinator and administrative team). It was unclear for two pilot trials whether changes would be made.

Four pilot trials reported definitive RCT funding intentions: two National Institute for Health Research (NIHR) Health Technology Assessment, one European and Developing Countries Clinical Trials Partnership, and one reported that a funding application had been prepared and submitted but did not specify the funder. One trial reported an anticipated progression time frame, specifying a recruitment start year for the definitive RCT.

### A priori progression criteria reporting of included randomised pilot trial publications

Trial protocols were available for 16 of the 37 (43%) completed randomised pilot trial publications (table 3). Trial registrations were identified for 20 of the trials without a published protocol. We were unable to identify a published protocol or trial registration for the one remaining completed pilot trial.

Twenty-two published protocols or trial registrations for the 37 included completed pilot trials did not report progression criteria. An additional four protocols or trial registrations reported different progression criteria to the pilot trial result publication. Only one completed trial publication explained why the progression criteria had changed from the protocol: as the qualitative findings were reported in a separate publication, the progression criteria associated with acceptability were not included in the completed pilot trial result publication.

### Progression criteria in prepublication peer reviewer reports

Prepublication peer reviewer reports were available for 153 of the 160 (96%) included external pilot publications. Peer reviewer reports were not publicly available for the three *PLoS One* publications and peer review was not commissioned for four of the pilot trial protocols published in *BMJ Open*.

Table 2 shows that over half of the prepublication peer reviewer reports commented on progression criteria (86/153, 56%). Peer reviewer reports for 35 pilot trial publications (6 completed, 29 protocol) indicated that progression criteria were not reported in the submitted prepublication manuscript. Whether progression criteria were reported in the submitted prepublication manuscript was unclear for another five pilot trial publications.

Peer reviewer reports for 20 of the publications referred to the rationale or justification given for progression criteria. For example, they asked why a specific progression criterion was set, why progression criteria were given for specific outcomes, how the progression criteria were established and how the progression decision was or will be made.

Peer reviewer reports for 26 pilot trial publications mentioned other aspects of progression criteria. For example, they mentioned changing where the progression criteria were reported in the manuscript (such as including the progression criteria in the publication abstract and not solely within a supplementary file), clarifying ambiguous wording, adding percentages in brackets for clarity, correcting inconsistencies in the manuscript and clarifying how specific criteria will be assessed. Reviewers also complemented authors for describing progression criteria well.

Not every author opted to update or add progression criteria to their manuscript after prepublication peer review. The authors of one publication argued that they could not alter their progression criteria because these criteria had been agreed by the trial management group, trial steering committee and ethics committee. Other reasons that authors gave in response to peer review for not reporting quantifiable numerical targets for progression criteria included: they were not set during trial design; strict thresholds might be influenced by contextual variations that may not affect a future trial; progression criteria are best viewed as guidelines in line with the CONSORT extension statement; different perspectives could not be successfully captured by a set of criteria; and the trial is not an internal pilot.

## DISCUSSION
### Summary of main findings

Our study provides an assessment of the reporting of progression criteria in a large sample of external randomised pilot trials. We found that progression criteria varied widely and were not often justified, which agrees with recent research assessing the use of progression criteria in internal pilot trials.[22][23] Like internal pilots, many of the studied external randomised pilot trials reported the intention to proceed to a definitive RCT when they had not strictly met all progression criteria, demonstrating flexibility in approach to progression decision-making with many opting to make changes to the definitive trial design. It was unclear within the studied publications how, or by whom, progression criteria are established and assessed.

Our findings suggest that guidance is needed to facilitate transparent and complete reporting of progression criteria a priori in external pilot trial protocols[3] and in pilot trial results publications.[24]

### Strengths and weaknesses of the study

A study strength is our extensive recent sample of 160 publications reporting external randomised pilot trials in four key journals that are known to publish pilot trials.

A limitation of this study is that single screening was used, and double data extraction was only conducted for 25% of the included publications. However, since only minimal data extraction differences were identified we decided not to conduct double data extraction for all included publications.

We only included external randomised pilot trials and it is unclear whether these findings are generalisable to other external feasibility study designs, such as non-randomised pilot trials and non-pilot feasibility studies. Our findings are also limited to four included journals and we did not include publications of non-English language which could introduce potential language bias.[25]

In addition, our review of peer reviewer reports to assess the context in which progression criteria were mentioned was subject to interpretation. Prepublication peer reviewer reports were not available for all included publications: *PLoS One* allows authors to opt in to publish peer reviewer reports, and peer review was not commissioned for four pilot trial protocols published in *BMJ Open* that had already been peer reviewed for ethical and funding approval before submission. Progression criteria might also have been added or altered based on editorial review before peer review. Unlike peer reviewer reports, it is not common practice to make editorial review publicly available.

### Meaning of the study: possible explanations and implications for clinicians and policymakers

Our findings suggest that the research community is uncertain about how progression criteria should be applied to external randomised pilot trials and how this should be reported in protocol and results publications. We identified one instance within a peer reviewer report for a pilot trial protocol where authors had stated that progression criteria were not set because the trial was not an internal pilot and as such would not immediately progress to a fully powered RCT.

We found recruitment and retention rates to be the most common feasibility uncertainties to contribute towards progression criteria. This result is supported by a recent

review finding recruitment to be the most common uncertainty evaluated in surgical pilot and feasibility studies,[26] and is unsurprising considering that recruiting to target is a challenge for many RCTs.[27] Fairhurst and colleagues suggested that researchers conducting feasibility studies might focus on feasibility uncertainties that are perceived to be important to funders.[26] In support of this suggestion, we found that other feasibility uncertainties that are equally as important to trial success, such as intervention acceptability, contributed to progression criteria much less often than recruitment and retention.

We found that peer review improved the reporting of pilot trials, for example, by prompting authors to explain their progression criteria and rationale. However, we also identified instances where new progression criteria were likely added as a result of peer review, in both protocols and pilot trial result manuscripts. Adding post hoc progression criteria could introduce bias since progression criteria might be based on targets that have been met or exceeded to justify progression to a definitive RCT.

### Unanswered questions and future research

The processes for establishing and assessing progression criteria were not commonly reported, leaving unanswered questions about how the decision to progress from pilot to definitive RCT is made in practice. This under-reporting could be due to a lack of guidance around best practices for progression of external randomised pilot trials, and how this should be reported in pilot trial publications. To expand on these findings a qualitative research study is being conducted to explore different stakeholders' perspectives and experiences of using progression criteria to inform the decision to progress from an external randomised pilot trial to a definitive RCT in practice.[28] Our findings also highlight the importance of journal editor and peer reviewer endorsement of evidence-based guidelines to improve reporting standards. The development of evidence-based guidance specific to the application and reporting of progression criteria in external pilot trials, for both protocols and completed trials, is a research priority. This finding is timely, as the UK's biggest funder of pilot and feasibility studies, NIHR Research for Patient Benefit, now stipulates that a clear route of progression (eg, progression criteria) should be included in pilot and feasibility study funding applications.[29] A further possible area of investigation is whether research ethics committees can and should comment on progression criteria in research ethics applications. Researchers have an ethical obligation to conduct research with integrity and transparency. Defining a priori progression criteria and adequately reporting them helps to uphold the integrity and transparency of the external randomised pilot trial's progression.

### CONCLUSIONS

We found heterogeneity in the reporting of progression criteria in external randomised pilot trial publications. It was often unclear how progression criteria were established, on what justification or rationale they were based,

how they were or will be assessed and who is involved at each stage. Peer reviewers often commented on progression criteria, questioning whether these criteria were established a priori, as is recommended for good practice. Clear, evidence-based recommendations for the use and reporting of progression criteria in external randomised pilot trials are required. Guidance to this effect would benefit researchers, peer reviewers, journal editors and funders of external randomised pilot trials, and inform the design of subsequent definitive trials. In the meantime, we suggest researchers consider reporting how their progression criteria were established in their pilot trial protocol publications, and how their findings in relation to progression criteria have informed progression decision-making and the subsequent definitive trial design in pilot trial results publications.

**Author affiliations**
¹Centre for Statistics in Medicine, Nuffield Department of Orthopaedics, Rheumatology and Musculoskeletal Sciences, University of Oxford, Oxford, UK
²Institute of Population Health Sciences, Barts and the London School of Medicine and Dentistry, Queen Mary University of London, London, UK
³Institute of Applied Health Sciences, University of Aberdeen, Aberdeen, UK
⁴School of Health and Related Research, The University of Sheffield, Sheffield, UK
⁵School of Medicine and Keele Clinical Trials Unit (CTU), Keele University, Keele, UK
⁶Department of Health Research Methods, Evidence and Impact, McMaster University, Hamilton, Ontario, Canada

**Acknowledgements** We acknowledge Dr Jennifer A de Beyer, Centre for Statistics in Medicine, University of Oxford, for English language editing.

**Contributors** This study was conceived and designed as part of KM's doctoral thesis under the supervision of SH, SD and SEI. CB, MJC, GL and LT provided feedback on the study design during protocol development. KM collected and analysed the data. SEd and NP conducted double data extraction. KM drafted the manuscript. All authors reviewed and commented on manuscript drafts. All authors approved the final manuscript.

**Funding** This work was supported by Medical Research Council Doctoral Training Partnership funding (grant number MR/N013468/1) awarded to KM.

**Competing interests** KM, SEd, CB, MJC, GL, LT, SEI and SH contribute to a Pilot and Feasibility Studies working group. GL and LT are editors in chief of the *Pilot and Feasibility Studies* journal. SEI, MJC and CB are editors of the *Pilot and Feasibility Studies* journal.

**Patient consent for publication** Not required.

**Ethics approval** Ethics committee approval is not required for this review since only previously published data are included.

**Provenance and peer review** Not commissioned; externally peer reviewed.

**Data availability statement** Data are available upon reasonable request. Data from this review will be included in a DPhil thesis published open access through the Oxford University Archive upon KM's DPhil completion.

and indication of whether changes were made. See: https://creativecommons.org/licenses/by/4.0/.

## ORCID iDs
Katie Mellor http://orcid.org/0000-0002-4054-5975
Michael J Campbell http://orcid.org/0000-0003-3529-2739

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
