## [Reviewer comments · BMJ Open]

ARTICLE DETAILS

TITLE (PROVISIONAL)	Progression from external pilot to definitive randomised controlled trial: A methodological review of progression criteria reporting
AUTHORS	Mellor, Katie; Eddy, Saskia; Peckham, Nicholas; Bond, Christine; Campbell, Michael; Lancaster, Gillian; Thabane, Lehana; Eldridge, Sandra; Dutton, Susan; Hopewell, Sally

VERSION 1 – REVIEW

REVIEWER	Young, Hannah University of Leicester, Department of Respiratory Sciences
REVIEW RETURNED	10-Mar-2021

GENERAL COMMENTS	Many thanks for asking me to review this interesting paper which aims to describe in detail the reporting of progression criteria, assess whether the progression criteria reported in pilot trial results publications were specified a priori, determine whether the intention to progress to a definitive RCT was reported in the results publication and assess the extent and context in which progression criteria were discussed in prepublication peer review reports where available. Overall, the paper is really well written, and I only have a few suggestions for amendment, which are as follows: • Throughout the results the use of descriptive statistics to quantify the findings is inconsistent. For example, where you write ‘Most of the pilot trial publications that reported the intention to conduct hypothesis testing stated that this was exploratory or advised caution in interpretation’ on page 7, can you quantify this. Similarly ‘Recruitment (113/160) and retention (106/160) were the most commonly reported indicators...’ on page 8. There are quite a few instances where the proportions have been omitted where it would be helpful for them to be included.• A slightly pedantic point, but on page 8 and page 13 you state “Recruitment (113/160) and retention (106/160) were the most commonly reported indicators of feasibility to inform progression criteria” and “NIHR Research for Patient Benefit, now stipulates that a clear route of progression criteria should be included in pilot and feasibility study funding applications” respectively. I am not sure this quite makes sense here, consider deleting ‘criteria’ in both instances.• Figure 1 – can you clarify what you mean by ‘non in human healthcare context’ within the figure. I also think that it is note worthy that around 37% of the abstracts screened were excluded because they did not report any progression criteria, which given these are recently published, further highlights the ongoing challenges with the use and reporting of progression criteria in pilot studies.
--

- Table 1. The mean target sample sizes for the protocol papers included seem very large for pilot studies – are these correct? The grey scale parts are a little tricky to read and it may be worth considering changing this here for readability.
- Some of the proportions within Tables 1,2 and 3 don't seem to add up. Please check.
- Page 9 you state “Although the findings from qualitative research conducted as part of a pilot trial are often reported in a separate publication, the intention to conduct qualitative research as part of a pilot trial should be included in the pilot trial protocol.” Do you have a reference for this statement? Is it drawn from guidance?
- Page 10 you describe the processes used for establishing progression criteria. Could you add information on the number of studies that have involved patient/ public representatives in the development of their progression criteria. Arguably these stakeholders have important views on progression criteria, particularly the acceptability of an intervention, and efforts to include patients in methodological decision making within trials is important. It would be useful to know how often this is achieved within recent work.
- Another pedantic point, but there is inconsistent use of numerals in the manuscript, for example, page 9 “Two protocols...”, page 10 “12 pilot trial publications” and “Forty-four of the 160 publications”. Please ensure consistency unless journal formatting guidance says otherwise.
- On page 12, summary of main findings, you state “Like internal pilots, many of the studied external randomised pilot trials also reported the intention to proceed to a definitive RCT even when they had not strictly met all progression criteria.” To my knowledge there is no stipulation that all progression criteria must be met to proceed to a full trial. Indeed, the traffic light system proposed by Avery et al allows for flexibility, with the ‘change’ threshold allowing researchers to identify where there are issues that may be remedied, rendering a definitive RCT viable. Consider rewording this section.
- The discussion could be further developed. You discuss uncertainty around progression criteria, the focus on criteria relating to recruitment and retention and the influence of peer review, but your original aims were much broader. The rationale and development of progression criteria was under-reported for example, and this warrants further discussion. This is likely due to a lack of specific guidance in this area, is there work underway or guidance updates that will address this issue? In the meantime, can you suggest what researchers should report or consider when developing their criteria which would be of benefit to researchers conducting pilot work.

Finally although a rationale has been provided, the focus on 4 key journals could be considered a limitation. The use of progression criteria may be very different across a broader selection of journals and also introduces an English language bias.

I hope that you find these helpful and many thanks again for inviting me to review this paper.

VERSION 1 – AUTHOR RESPONSE

Reviewer: 1

Dr. Hannah Young, University of Leicester, University Hospitals of Leicester NHS Trust

Comments to the Author:

Many thanks for asking me to review this interesting paper which aims to describe in detail the reporting of progression criteria, assess whether the progression criteria reported in pilot trial results publications were specified a priori, determine whether the intention to progress to a definitive RCT was reported in the results publication and assess the extent and context in which progression criteria were discussed in prepublication peer review reports where available.

Overall, the paper is really well written, and I only have a few suggestions for amendment, which are as follows:

- Throughout the results the use of descriptive statistics to quantify the findings is inconsistent. For example, where you write ‘Most of the pilot trial publications that reported the intention to conduct hypothesis testing stated that this was exploratory or advised caution in interpretation’ on page 7, can you quantify this. Similarly ‘Recruitment (113/160) and retention (106/160) were the most commonly reported indicators...’ on page 8. There are quite a few instances where the proportions have been omitted where it would be helpful for them to be included.

Author: Thank you for your comment. We have checked all descriptive statistics are clear and have added percent proportions where they were not previously reported.

- A slightly pedantic point, but on page 8 and page 13 you state “Recruitment (113/160) and retention (106/160) were the most commonly reported indicators of feasibility to inform progression criteria” and “NIHR Research for Patient Benefit, now stipulates that a clear route of progression criteria should be included in pilot and feasibility study funding applications” respectively. I am not sure this quite makes sense here, consider deleting ‘criteria’ in both instances.

Author: Thank you for this suggestion, we have deleted the word ‘criteria’ on page 9. On page 13 we have amended to state that “a clear route of progression (e.g., progression criteria) should be included...”

- Figure 1 – can you clarify what you mean by ‘non in human healthcare context’ within the figure. I also think that it is note worthy that around 37% of the abstracts screened were excluded because they did not report any progression criteria, which given these are recently published, further highlights the ongoing challenges with the use and reporting of progression criteria in pilot studies.

Author: Thank you for your comment. We have clarified the reasons for exclusion in the figure and have re-ordered reasons for exclusion e.g., internal pilot exclusion should also come under non-eligible design. We have also acknowledged under *results – study selection* that we excluded many publications at full text stage because we could not identify explicit progression criteria reported in the publication.

• Table 1. The mean target sample sizes for the protocol papers included seem very large for pilot studies – are these correct? The grey scale parts are a little tricky to read and it may be worth considering changing this here for readability.

Author: Thank you for your comment. We can confirm that the sample size targets are correct. There were a couple of included cluster randomised pilot trial protocol publications with very large sample size targets, one of 12,000 (1) and another of 6000 (2) which have skewed the mean. Although this research does not focus on the issues around external pilot trial sample sizes, one of the co authors is conducting doctoral research in this area.

- (1) McIntyre L, Taljaard M, McArdle T, et al. FLUID trial: a protocol for a hospital-wide open-label cluster crossover pragmatic comparative effectiveness randomised pilot trial. *BMJ Open*. 2018;8(8):e022780. Published 2018 Aug 23. doi:10.1136/bmjopen-2018-022780
- (2) van Oostveen RB, Romero-Palacios A, Whitlock R, et al. Prevention of Infections in Cardiac Surgery study (PICS): study protocol for a pragmatic cluster-randomized factorial crossover pilot trial [published correction appears in *Trials*. 2019 Oct 16;20(1):595]. *Trials*. 2018;19(1):688. Published 2018 Dec 17. doi:10.1186/s13063-018-3080-y

We have also formatted all table text in black for readability.

• Some of the proportions within Tables 1,2 and 3 don't seem to add up. Please check.

Author: Thank you, we have checked the proportions. All percent proportions are given to two significant figures and are of the publication type (completed, protocol or total). We have added a footnote to all tables to state that percentages may not sum to 100 due to rounding.

• Page 9 you state “Although the findings from qualitative research conducted as part of a pilot trial are often reported in a separate publication, the intention to conduct qualitative research as part of a pilot trial should be included in the pilot trial protocol.” Do you have a reference for this statement? Is it drawn from guidance?

Author: Thank you for this comment. We have added a reference to Cooper et al 2014 where the following recommendation is made – *if qualitative research will be used to adapt, amend or refine either the intervention or aspects of trial conduct during the trial... this should be made explicit and the processes by which it will occur should be clear before the trial commences.*

We acknowledge that some researchers might document qualitative research protocols separately from the main pilot trial protocol but we would still expect there to be reference to the intention to conduct qualitative research in published pilot trial protocols. We have updated the wording to reflect that “*the intention to conduct qualitative research as part of a pilot trial should be made explicit before the pilot trial commences and be included in the pilot trial protocol*”.

• Page 10 you describe the processes used for establishing progression criteria. Could you add information on the number of studies that have involved patient/ public representatives in the development of their progression criteria. Arguably these stakeholders have important views on progression criteria, particularly the acceptability of an intervention, and efforts to include patients in

methodological decision making within trials is important. It would be useful to know how often this is achieved within recent work.

Author: Thank you for this comment. We agree that including patient and public representatives in developing progression criteria is an important consideration and was not something that was reported explicitly, except for in one publication that stated that progression criteria were “*specified by the Trial Management Group (TMG), including the PPI representatives*”. Therefore we have looked at the studies that reported who was involved in developing progression criteria and where a Trial Steering Committee or Trial management Group were involved we have reported how many of these had Patient and Public contributors.

- Another pedantic point, but there is inconsistent use of numerals in the manuscript, for example, page 9 “Two protocols...”, page 10 “12 pilot trial publications” and “Forty-four of the 160 publications”. Please ensure consistency unless journal formatting guidance says otherwise.

Author: Thank you, we have updated throughout for consistency (numbers 1-10 spelled out, numbers >10 written as numerals except when at the start of a sentence).

- On page 12, summary of main findings, you state “Like internal pilots, many of the studied external randomised pilot trials also reported the intention to proceed to a definitive RCT even when they had not strictly met all progression criteria.” To my knowledge there is no stipulation that all progression criteria must be met to proceed to a full trial. Indeed, the traffic light system proposed by Avery et al allows for flexibility, with the ‘change’ threshold allowing researchers to identify where there are issues that may be remedied, rendering a definitive RCT viable. Consider rewording this section.

Author: Thank you for this comment. We agree and can see how the current wording does not account for this flexibility in approach to progression decision making based on progression criteria and pilot trial findings. We have amended this paragraph.

- The discussion could be further developed. You discuss uncertainty around progression criteria, the focus on criteria relating to recruitment and retention and the influence of peer review, but your original aims were much broader. The rationale and development of progression criteria was under-reported for example, and this warrants further discussion. This is likely due to a lack of specific guidance in this area, is there work underway or guidance updates that will address this issue? In the meantime, can you suggest what researchers should report or consider when developing their criteria which would be of benefit to researchers conducting pilot work.

Author: Thank you for this suggestion, we agree that this is an important point and have added further details of future research we are conducting with reference to a protocol for a qualitative research study that is currently underway. We have also updated the conclusion with the suggestion that researchers report how criteria are established a priori and how they subsequently inform progression decision-making.

Finally although a rationale has been provided, the focus on 4 key journals could be considered a limitation. The use of progression criteria may be very different across a broader selection of journals and also introduces an English language bias.

Author: Thank you. We agree that this could also be a potential limitation of the review and have acknowledged this in the discussion.

VERSION 2 – REVIEW

REVIEWER	Young, Hannah University of Leicester, Department of Respiratory Sciences
REVIEW RETURNED	28-May-2021
GENERAL COMMENTS	The authors have fully addressed all my previous comments, thank you.